

# Development and initial validation of the Psychological Need Frustration Scale for Physical Activity

Pak-Kwong Chung[1], Tao Zhong[2], Jing-Dong Liu[3], Chun-Qing Zhang[1] and Ming Yu Claudia Wong[1]

[1] Department of Sport and Physical Education, Hong Kong Baptist University, Hong Kong, China
[2] College of Sport and Health, Henan Normal University, Henan, China
[3] Department of Physical Education, Sun Yat-Sen University, Guangdong, China

## ABSTRACT

**Background**. The frustration of basic psychological needs can be detrimental to people's health. To date, a scale developed specifically for measuring such perceived negative experiences, derived from a need thwarting environment in the physical activity context, is lacking. The present research attempted to develop and validate the Psychological Need Frustration Scale for Physical Activity (PNFS-PA) grounded in self-determination theory via multiple studies.

**Method**. In Study 1, an item pool was created, and its face and content validity were established. In Study 2, the factor structure of the scale was demonstrated using exploratory structural equation modelling (ESEM). In Study 3, its factor structure was cross-validated. Also, the nomological validity, reliability and measurement invariance of the scale were established.

**Result**. Taken together, the research suggests the newly developed PNFS-PA is valid and reliable and can be applied to assess psychological needs frustration experiences in the physical activity context.

## INTRODUCTION

Engaging in physical activity (PA) regularly is an effective way to maintain a healthy and fit body (*Lee et al., 2012*; *Feeny et al., 2014*; *Humphreys, McLeod & Ruseski, 2014*). However, physical inactivity remains a major problem in global health (*Pratt et al., 2014*; *Trost, Blair & Khan, 2014*). As such, facilitating PA among the general population is an important task in health and fitness promotion initiatives. In this sense, basic psychological needs theory (BPNT), as a sub-theory in self-determination theory (SDT; *Ryan & Deci, 2017*), might be a useful framework to help understand PA behavior. According to BPNT, there are three innate, nonhierarchical and universal psychological needs, namely needs for autonomy, competence and relatedness (*Ryan, 1995*). Autonomy is defined as a sense of self-volition and self-governance over one's behavior. Competence refers to the perception that one can

Corresponding authors
Pak-Kwong Chung,
pkchung@hkbu.edu.hk
Tao Zhong,
tzhong_research@hotmail.com

successfully achieve a challenging task. Relatedness is characterized as perceived positive and meaningful connections with others, and the feeling of belonging (*Ryan & Deci, 2017*).

To date, a large number of studies have revealed that people with high need satisfaction likely experience self-determination for behavior (e.g., PA behavior), optimal functioning, personal growth and subjective well-being (*Reeve & Jang, 2006*; *Williams et al., 2009*). Despite the relative plethora of research on needs satisfaction, only a few recent attempts have been made to explore the influence of need frustration. As advocated by *Ryan & Deci (2017)*, not only the bright side (needs satisfaction) but also the dark side (needs frustration) of human experiences should be taken into account. Need frustration is referred to as perception of the basic psychological needs are being thwarted, which is undermining, alienating and pathogenic (*Ryan & Deci, 2017*). Specifically, autonomy need frustration can occur in an environment characterized as controlling. Competence need frustration can take place when people are made to feel ineffective in an environment demeaning of their ability. Relatedness need frustration can happen in a cold and neglectful environment (*Vansteenkiste, Niemiec & Soenens, 2010*). As a distinct psychological aspect, need frustration is not equivalent simply with the state of lacking basic psychological need satisfaction (*Bartholomew et al., 2011*). As demonstrated in previous empirical studies, lack of need satisfaction is not ideal for negative outcomes and maladaptation prediction (*Adie, Duda & Ntoumanis, 2012*; *Mack et al., 2012*). Therefore, previous measures of basic psychological needs satisfaction are inappropriate to measure need frustration experiences, as their items only capture positive aspects of basic psychological needs. Developing measures with items that explicitly tap into the negative aspects of basic psychological needs (i.e., need frustration) is warranted.It is posited that negative consequence (e.g., fragmentation, maladaptation and ill-being) would be more salient when basic psychological needs frustration are directly measured (*Bartholomew et al., 2011*; *Vansteenkiste & Ryan, 2013*).

In the context of PA, people could experience need frustration when engaging in PA in a social environment characterized as need thwarting (*Ng et al., 2013*). Thwarting behavior from others (e.g., family members, friends, co-workers, classmates, PA instructors) could result in need frustration. For example, if a husband uses contingent rewards to motivate his wife to participate in PA, this may frustrate her basic psychological needs and make her feel pressured. To explore need frustration experiences in the PA specific context, a psychometrically sound measurement is desirable. However, currently, there is no such measurement available for researchers, which limits investigation in this domain. Although an endeavor has been made to modify a need frustration scale from a competitive sport context (*Bartholomew et al., 2011*; *Martinent, Guillet-Descas & Moiret, 2015*) to a PA context (*Gunnell et al., 2013*), such modification is somewhat problematic. For instance, the values of the CFI and TLI resided in an acceptable range (i.e., above .90), but the value of the RMSEA (.10) was not acceptable (i.e., below .08; *Hu & Bentler, 1999*), indicating the model may not fit the data very well. Further, an inspection of the content of the modified scale indicated that some modified items may not fit the PA context. For instance, the item "I feel inadequate because I am not given opportunities to fulfil my potential" in the competence need frustration factor and the item "I feel that other people are envious

when I achieve success" in the relatedness need frustration factor is more pertinent to athletes in the context of a competitive sport rather than the PA context. PA behaviors of the general population are more health maintenance/enhancement oriented other than competition-oriented, and descriptions such as "fulfilling potential" and "achieving success" are not very relevant to the PA context. Taken together, it appears that the modified scale from competitive sport may not be an ideal instrument for measuring PA need frustration and there is an opportunity to expand and improve upon prior work in this area.

Besides, traditionally confirmatory factor analysis (CFA) was employed in validations studies of need frustration scales (*Gunnell et al., 2013*). However, it has been criticized for relying on a highly restrictive independent cluster model, where cross-loadings of items on unintended factors are forced to be zero (*Asparouhov & Muthén, 2009*). Given the three basic psychological needs are distinct while inter-correlated, CFA approach might be inappropriate in analyzing the structure of their measurement. Also, a highly restrictive model in CFA would inflate the correlations of latent factors, which could fail to detect the genuine correlations between factors (*Joshanloo, Bobowik & Basabe, 2016*). For instance, inflated factor correlations have been found in prior studies using CFA, where the correlations between autonomy need frustration and competence need frustration, between autonomy need frustration and relatedness need frustration, and competence need frustration and relatedness need frustration were .96, .85 and .95, respectively (*Gunnell et al., 2013*). To overcome the limitations from a statistical perspective, exploratory structural equation modelling (ESEM) has been advocated and gaining its popularity. ESEM allows for non-significant cross-loadings, which enables researchers to freely estimate cross-loadings and it has less restrictive assumptions (*Marsh et al., 2014*). Moreover, it is an advance to traditional exploratory factor analysis (EFA), as it integrates the principles of the EFA into the framework of SEM, which allows for the examination of more complex models, such as measurement invariance models in validation studies. Therefore, ESSM could be adopted as a statistical approach in the present research.

The purpose of this research was to develop and validate the psychological need frustration scale for PA (PNFS-PA). Three studies were included to achieve the research purpose. In Study 1, the item pool of the PNFS-PA was produced, and the face and content validity of items were examined. In Study 2, the initial PNFS-PA produced in Study 1 was examined to explore scale structure through ESEM. In Study 3, the factor structure was cross-validated in an independent sample via ESEM. Additionally, nomological validity, reliabilities (Cronbach's alpha, composite reliability and test-retest reliability) and measurement equivalence of the scale were examined.

## METHODS AND RESULTS

### Study 1

The purpose of this study was three-fold. First, it was to generate an item pool of the PNFS-PA, which was inputted from both the relevant measurements and qualitative interviews. Second, it aimed to assess the PNFS-PA's face validity through participants

drawn from the targeted population. Third, the PNFS-PA's content validity reflecting the extent to which an instrument measures the underlining theoretical construct was evaluated by specialists in the research field.

## Method
### *Participants*

Nineteen Chinese participants (age = 38.68 ± 14.13, age-range 20–64, nine males and ten females) were recruited and invited for interviews, through which their need frustration experiences under the PA context was explored. Inclusion criteria were: general Chinese adults in Hong Kong aged between 18 and 64. Exclusion criteria were: adults with major complaints or medical conditions that may interfere with PA engagement; athletes and students majored in PE/sport; and manual workers or people whose work requires a high PA level.

In addition, after the initial item pool for the PNFS-PA was formed, ten Chinese adults (age = 43.90 ± 15.35, age-range 21–64, five males and five females) were invited to assess its face validity. Recruitment criteria were the same. Moreover, three Chinese sport and exercise psychologists with expertise in SDT were invited to review the content validity of the items of the PNFS-PA.

### *Procedure*

Prior to commencement, the study was approved by the Research Ethics Committee at Hong Kong Baptist University (HASC/16-17/0304). An initial item pool was sourced from two inputs. First, items from need frustration measurements in other related domains such as competitive sport (*Bartholomew et al., 2011*) and physical education (*Liu & Chung, 2015*) were referred. Items were carefully reviewed to examine their applicability to the PA context. Those that could benefit the creation of the PNFS-PA's item pool were borrowed and adapted. Second, interviews on participants' need frustration experiences in the PA context were undertaken to supplement the PNFS-PA's item pool. Following a semi-structured interview guideline, participants were provided with lay definitions of the basic three psychological needs and asked by considering their (past and/or present) PA experiences, to discuss situations in which their feelings of autonomy, competence, and relatedness were undermined. Subsequently, the PNFS-PA's item pool was presented to participants so as to check the face validity, readability, clarity and comprehensibility of items. Candidate items perceived as inapplicable/redundant by 70% or more of the participants were deleted.

The participants in study 1 were recruited using convenience sampling. They were recruited from several local communities (university campuses and public spaces such as squares and parks) in Hong Kong, via the distribution of invitation sheets. Interested participants consented and engaged in the study.

To assess the content validity of the PNFS-PA's pool of items, three Chinese specialists from universities in Hong Kong were contacted via email. They received their PhD training in (sport and exercise) psychology and had SDT-related publications in international peer-refereed journals. The three judges were provided with a definition of the psychological need frustration and were invited to review to what extent the items were applicable to the
PA context. They rated individual items on a four-point Likert scale (1 = not applicable; 2 = somewhat applicable; 3 = quite applicable; 4 = highly applicable) using their expertise. Subsequently, the content validity index (CVI) was calculated for every single item for further consideration (whether a certain item would be retained or deleted). The CVI of each item was computed by dividing the number of experts who gave a rating on either "3" or "4" by three, the total number of experts involved.Consistent with previous research, a value of CVI = 1.00 was considered excellent, while a CVI's value of 0.67 was deemed acceptable. A CVI of 0.33 or less resulted in the corresponding deletion of a given item (*Polit & Beck, 2006*). CVI is an important approach to evaluating content validity by quantifying the degree to which elements of an assessment instrument are relevant to a targeted construct (*Almanasreh, Moles & Chen, 2019*). Its use can be seen in research developing psychological need frustration instrument for competitive sport context (*Bartholomew et al., 2011*) and physical education context (*Liu & Chung, 2015*). In addition, suggestions concerning any possible ambiguity and revisions of candidate items were also derived from the experts.

Of note, the term "physical activity" used in the research refers to any bodily movement produced by skeletal muscles that require energy expenditure (*Caspersen, Powell & Christenson, 1985*). To facilitate the term understanding by participants involved in the research, some common forms of PA such as running, dancing, swimming, yoga and other recreational sports (basketball, badminton, etc.) were provided. The term provides the context in which a person may experience need frustration, such as from physical activity instructors, significant others and peers who do not understand and appreciate PA-related decisions and perspectives.

## RESULTS

A total of 34 items (in Chinese) were initially generated for the PNFS-PA. During the face validity evaluation phase, 15 items that were classified as inapplicable or redundant with other items based on participants' feedback were deleted, resulting in 19 items. In addition, the relevance, comprehensibility and clarity of the remained items were checked by the participants. The items retained were successively reviewed by three SDT experts. Based on their feedback, three items (two items in competence need frustration factor and one item in relatedness need frustration factor) were rated as inapplicable, namely scored as "1" or "2" by two judges, with a value of CVI of 0.33. Therefore, they were deleted which remained 16 items. Also, qualitative feedback regarding the further improvement of items was elicited, and minor revisions were made to enhance the quality of the scale's items. In summary, after following serial steps of PNFS-PA's item pool creation and refinement, an initial 16-item scale (six items for autonomy need frustration, four items for competence need frustration and six items for relatedness need frustration) was formed for subsequent studies.

### Study 2
The purpose of Study 2 was to examine the factor composition of the initial scale via exploratory structural equation modelling (ESEM) using Mplus version 7.0. The
examination was to: (a) avoid misspecification of items in each underlying factor (e.g., ascertain item created for the competence need frustration load on the competence need frustration factor); (b) maximize the convergent and discriminant validity of the scale.

## Method

### Participants and procedure

Participants were three hundred and thirty Chinese adults in Hong Kong. Their age ranged from 18 to 63 ($M = 33.85$, SD $= 15.12$), with 141 males and 189 females. Inclusion/exclusion criteria were the same as that of the last study. An invitation sheet containing the study information and an online survey QR code was created and distributed to university students in Hong Kong, to invite them and people in their close social environment (e.g., relatives, friends, etc. and cover the age range 18–64) to participate in the study. Interested participants signed the informed consents online and engaged in the study. In the introduction section of the online survey, the term "physical activity" was explained, as did in study 1. This could help participants better understand the context in which need frustration experience could take place. Participants were also clearly informed regarding the voluntary and anonymous nature of their participation at the beginning of the online survey. Research Ethics Committee at Hong Kong Baptist University approved the conduct of the study.

### Measures

*Demographic information.* For basic demographic information, two items concerning participants' age and gender were asked in the survey battery.

*Need frustration.* The 16-item PNFS-PA generated from study 1 was adopted. Items are anchored along a seven-point Likert scale ranging from 1 (strongly disagree) to 7 (strongly agree).

## Data analysis

ESEM was conducted to explore the structure of the PNFS-PA using Mplus Version 7.0 (*Muthén & Muthén, 2015*). Oblique geomin rotation was used with an epsilon value of 0.5 and a robust maximum likelihood (MLR) estimation. Three factors were specified in the model for the autonomy, competence and relatedness need frustration, respectively. Given $\chi^2$ issensitive to sample size, an array of indices including comparative fit index (CFI), Tucker-Lewis index (TLI), root mean square error of approximation (RMSEA), and standardized root mean square residual (SRMR) was used to evaluate the model fit (*Hu & Bentler, 1999*). In line with previous research using ESEM (*Marsh, Hau & Grayson, 2005*), CFI and TLI values >0.90, and RMSEA and SRMR values <0.08 are indicative of acceptable model fit, while CFI and TLI values >0.95, and RMSEA and SRMR values <0.06 are indicative of good model fit. When it comes to item retention/removal, the following criteria were employed: first, items with a large standardized factor loading (i.e., >.40) on unintended factor loading (e.g., autonomy need frustration item loaded highly on competence or relatedness need frustration factor) were removed; second, items

with a high cross-loading (i.e., >.30) with another factor were removed; third, items with a standardized primary factor loading <.40 (which manifests item did not load on any factor) were removed (*Matsunaga, 2010*).

## RESULTS

The initial ESEM displayed an adequate model fit to the data, where $\chi^2$ (75) = 178.334, < .001, CFI = .961, TLI = .938, RMSEA (90% CI) = .065 (.052, .077), SRMR = .024. However, an inspection of the standardized factor loadings based on aforementioned criteria revealed that one item in the autonomy need frustration factor and one item in the relatedness need frustration factor had a cross-loading problem, thus they were discarded and the second round of ESEM was implemented. The second round of ESEM resulted in an improved model fit, where $\chi^2$ (52) = 114.012, $p$ < .001, CFI = .973, TLI = .952, RMSEA (90% CI) = .060 (.045, .075), SRMR = .020. The standardized factor loadings based on the aforementioned criteria were inspected and found no problem. Therefore, the factor structure of the scale was initially established. As for factor labelling, according to item content, factor 1, 2 and 3 were labelled as autonomy, competence and relatedness need frustration, respectively. Coefficients of Cronbach's alpha of the autonomy, competence and relatedness need frustration subscales were .903, .881, and 940. The descriptive analysis result, standardized factor loadings of the scale's items and factor correlations are presented in Table 1. In summary, via the implementation of the ESEM, the factorial validity of the PNFS-PA was examined. Two items that showed poor discriminant validity were discarded, resulting in a scale with 14 items. The resultant scale would be administered to an independent sample in the next study to further investigate its validity and reliability.

### Study 3

The purposes of Study 3 were to: cross-validate the factor composition of the PNFS-PA via ESEM using Mplus version 7.0; establish the nomological validity of the scale through correlation analysis between the PNFS-PA scale's factors and related constructs based on the premise of SDT (i.e., subjective vitality and negative affect), and examine the measurement invariance of the newly designed scale.

### Method
#### Participants and procedure

As with the last two studies, participants were recruited in Hong Kong. Criteria for inclusion/exclusion were the same as study 1 and 2. Three hundred and twenty-two Chinese participants (age range 18–62, age = 37.44 ± 15.21 years, 158 males and 164 females) completed an online survey. In addition, an independent sample of 49 participants (age = 19.51 ± 1.21 years, age range 18–22, 21 males and 28 females) were invited to fill in the PNFS-PA twice (three week-interval between test and retest) to examine the scale's temporal stability.

The study procedure was the same as outlined in the last study. Informed consents from all participants were obtained online. Participants were clearly informed regarding

**Table 1  Item Content, Descriptive Analysis, Factor Loadings and Factor Correlations in Study 2 ($N = 330$).**

| Item Content | Mean | SD | SK | KU | AU | CO | RE |
|---|---|---|---|---|---|---|---|
| (Autonomy) When engaging in physical activity, at times you feel: | | | | | | | |
| Restricted from making choice | 3.485 | 1.558 | .242 | −.642 | .739 | .036 | .038 |
| Forced to follow decisions made for you | 3.282 | 1.572 | .273 | −.794 | .841 | −.001 | .033 |
| Other people make their demand without providing rationale | 3.194 | 1.573 | .441 | −.481 | .754 | .155 | .039 |
| Other people use excessive personal control | 3.039 | 1.478 | .517 | −.323 | .743 | .109 | .145 |
| Forced to do things that you don't want to | 2.979 | 1.554 | .568 | −.507 | .568 | .202 | .215 |
| (Competence) When engaging in physical activity, at times: | | | | | | | |
| You are made to feel powerless in some situations | 4.191 | 1.635 | −.197 | −.743 | .093 | .662 | -.006 |
| You feel incompetent because of things you are told | 3.924 | 1.582 | −.139 | −.791 | .105 | .750 | .069 |
| You doubt if you can achieve improvement because of comments you receive | 4.100 | 1.553 | −.207 | −.686 | .034 | .821 | .085 |
| You doubt your ability to overcome challenges because of comments you receive | 3.870 | 1.543 | −.005 | −.736 | .032 | .766 | .138 |
| (Relatedness) When engaging in physical activity, at times you feel: | | | | | | | |
| You are rejected by those around you | 2.818 | 1.555 | .550 | −.605 | .113 | .151 | .764 |
| Other people overlook you on purpose | 2.724 | 1.456 | .560 | −.423 | .100 | .088 | .764 |
| Other people say bad words about you | 2.600 | 1.485 | .754 | −.247 | .048 | .065 | .835 |
| Other people do not listen to you | 2.852 | 1.471 | .512 | −.595 | .071 | .175 | .755 |
| Other people are reluctant to offer help | 2.661 | 1.438 | .641 | −.438 | .118 | .113 | .766 |

| | CO | | | | | | RE |
|---|---|---|---|---|---|---|---|
| AU | .346[***] | | | | | | .352[***] |
| CO | | | | | | | .410[***] |

**Notes.**

SD, Standard deviation; SK, Skewness; KU, Kurtosis; AU, autonomy need frustration; CO, competence need frustration; RE, relatedness need frustration.

[***] $p < .001$

the voluntary and anonymous nature of their participation at the beginning of the online survey. Ethical clearance for the present study was sought and received from the authors' local research ethics committee.

### Measures

*Demographic information.* For basic demographic information, two items concerning participants' age and gender were asked in the survey battery.

*Need frustration.* The PNFS-PA established in Study 2 was adopted. It comprises 14 items (five items for the autonomy need frustration, four items for the competence need frustration and five items for the relatedness need frustration, respectively), anchored along a seven-point Likert scale, ranging from 1 (strongly disagree) to 7 (strongly agree).

*Subjective well-being.* The six-item Subjective Vitality Scale (SVS; *Bostic, Rubio & Hood, 2000*; *Ryan & Frederick, 1997*) that measures respondents' perception of positive energy
was used for nomological validity assessment. Responses are rated on a seven-point Likert scale ranging from 1 (strongly disagree) to 7 (strongly agree). The previous study has supported its application among the Chinese population (*Liu & Chung, 2015*).

*Negative affect.* The five-item negative affect factor from the ten-item International Positive and Negative Affect Schedule Short Form (IPANAS-SF; *Thompson, 2007*) was used to measure participants' negative affect. It served as the assessment of the nomological validity of the PNFS-PA. Respondents were asked to respond to items anchored along a five-point Likert scale ranging from 1 (never) to 5 (always). Previous research has supported its validity and reliability among the Chinese population (*Liu & Chung, 2015*).

### Data analysis

The strategy for factorial validity examination was the same as that of the last study. ESEM was conducted to cross-validate the factor structure of the PNFS-PA using Mplus Version 7.0 (*Muthén & Muthén, 2015*). With regard to the nomological validity assessment of the scale, based on the theoretical tenet of SDT (*Vansteenkiste & Ryan, 2013*; *Ryan & Deci, 2017*), it was predicted that need frustration would positively relate with negative affect, while negatively related with subjective well-being. As for the scale's reliability appraisal, once its structure was validated, internal consistency reliability assessment using Cronbach's alpha and composite reliability (*Raykov, 1997*) would be adopted. Additionally, the intra-class correlation and its 95% CI obtained from a two-way random model (*McGraw & Wong, 1996*) were adopted to assess three-week interval temporal stability of the scale.

Multi-group ESEM was used for measurement invariance testing across gender, age and independent samples. Samples from study 2 and 3 were used. To compare models and examine their invariance, since the $\chi^2$ difference test is dependent on sample size (*Marsh, Balla & McDonald, 1988*), it was not used. Instead, we used multiple fit indices containing CFI, RMSEA and SRMR. Changes less than .010 for CFI, .015 for RMSEA and .030 for SRMR are the criteria for measurement invariance (*Chen, 2007*). Moreover, information criteria consisting of the Akaike information criteria (AIC), the Bayesian information criterion (BIC), and the sample size adjusted BIC (ABIC) were also used for comparing models. Lower values of the information criteria indicate a better model fit to the data.

## RESULTS

The cross-validated model revealed an adequate goodness-of-fit to the data, where $\chi^2$ (52) = 122.674, $p < .001$, CFI = .969, TLI = .947, RMSEA = .065 (.050, .080), SRMR = .025. Table 2 shows the descriptive analysis results, standardized factor loadings of items and the factor correlations. As to the result of nomological validity examination, as displayed in Table 3, the correlations between scores of the autonomy, competence and relatedness need frustration sub-scales and the score of subjective vitality were in the expected negative direction, while the correlations between scores of the autonomy, competence and relatedness need frustration sub-scales and the score of negative affect were in the expected positive direction. Also, need frustration demonstrated a stronger

**Table 2   Item Content, Descriptive Analysis and Factor Loadings and Factor Correlations in Study 3 ($N = 322$).**

| Item Content | Mean | SD | SK | KU | AU | CO | RE |
|---|---|---|---|---|---|---|---|
| (Autonomy) When engaging in physical activity, at times you feel: | | | | | | | |
| Restricted from making choice | 3.677 | 1.631 | .136 | −.908 | .701 | .171 | -.037 |
| Forced to follow decisions made for you | 3.429 | 1.611 | .199 | −.737 | .727 | .170 | .085 |
| Other people make their demand without providing rationale | 3.320 | 1.571 | .309 | −.600 | .735 | .080 | .055 |
| Other people use excessive personal control | 3.115 | 1.583 | .511 | −.475 | .761 | .087 | .172 |
| Forced to do things that you don't want to | 3.118 | 1.613 | .457 | −.672 | .702 | .131 | .208 |
| (Competence) When engaging in physical activity, at times: | | | | | | | |
| You are made to feel powerless in some situations | 4.165 | 1.557 | −.202 | −.700 | .095 | .427 | .105 |
| You feel incompetent because of things you are told | 4.016 | 1.525 | −.106 | −.624 | .098 | .642 | .073 |
| You doubt if you can achieve improvement because of comments you receive | 4.006 | 1.551 | −.177 | −.633 | .035 | .901 | -.005 |
| You doubt your ability to overcome challenges because of comments you receive | 3.801 | 1.482 | −.012 | −.701 | .080 | .781 | .076 |
| (Relatedness) When engaging in physical activity, at times you feel: | | | | | | | |
| You are rejected by those around you | 2.814 | 1.542 | .659 | −.303 | .089 | .090 | .809 |
| Other people overlook you on purpose | 2.835 | 1.571 | .658 | −.417 | .060 | .123 | .834 |
| Other people say bad words about you | 2.689 | 1.505 | .744 | −.083 | .144 | .037 | .786 |
| Other people do not listen to you | 2.953 | 1.527 | .518 | −.577 | .100 | .139 | .773 |
| Other people are reluctant to offer help | 2.727 | 1.466 | .816 | .223 | .142 | .081 | .691 |

| | CO | RE |
|---|---|---|
| AU | .429[***] | .401[***] |
| CO | | .333[***] |

Notes.

SD, Standard deviation; SK, Skewness; KU, Kurtosis; AU, autonomy need frustration; CO, competence need frustration; RE, relatedness need frustration.

[***] $p < .001$

**Table 3   Cronbach's alpha and correlations for nomological validity examination in study 3 ($N = 322$).**

| | Subjective vitality ($\alpha = .915$) | Negative affect ($\alpha = .792$) |
|---|---|---|
| Autonomy need frustration ($\alpha = .919$) | −.105 | .207[***] |
| Competence need frustration ($\alpha = .840$) | -.136[*] | .182[**] |
| Relatedness need frustration ($\alpha = .938$) | -.141[*] | .291[***] |

Notes.

[*] $p < .05$.

[**] $p < .01$.

[***] $p < .001$.

correlation with the indicator of negative experiences than the indicator of positive experiences. The results confirmed the nomological validity of the newly developed scale.

The Cronbach's alpha coefficients of the three need frustration subscales were all above .800 (See Table 3), indicating the reliability of the newly designed scale was reasonable. Furthermore, as for the temporal stability assessment of the scale, the intra-class correlation

coefficients and their 95% CI for autonomy, competence and relatedness need frustration were .825 (.690, .901), .804 (.652, .890), and .865 (.761, .924), respectively. All were statistically significant at $p < .001$. The result showed the scale was stable over time.

Multi-group ESEM analysis results are displayed in Table 4. Based on goodness-of-fit indices, both the independent models and invariance models showed sufficient fit to the data. When comparing the models constrained to varying degrees, changes in model fit indices (i.e., CFI, RMSEA and SRMR) were all smaller than suggested cutoff values indicative of measurement invariance. Furthermore, information criteria (i.e., AIC, BIC and ABIC) showed consistent values in different comparing models. The result supported the scale was invariant across gender (male vs female), age groups, and independent samples, which indicated the comparison of scores of the PNFS-PA across groups would be meaningful.

# DISCUSSION

Basic psychological need frustration is detrimental to people's health (*Vansteenkiste & Ryan, 2013*), and can induce mal-adaptation and subjective ill-being (*Adie, Duda & Ntoumanis, 2012*; *Balaguer et al., 2012*). Given its diminishing impact, researchers have taken action to tap into such perceived negative experiences in various contexts such as competitive sport (*Bartholomew et al., 2011*) and physical education (*Van den Berghe et al., 2015*). However, need frustration, derived from a need thwarting environment and taking place in the PA context, has been less explored. Among all obstacles, lack of a valid and reliable measure is an important one. Therefore, this research filled the gap by developing the PNFS-PA and examining its psychometric properties. Overall, the newly designed 14-item scale revealed satisfactory psychometric characteristics, indicating it can be a useful tool for future research in the PA specific context.

A series of analysis procedures were followed to develop and validate the PNFS-PA. First, an item pool of the scale was generated from two inputs, that is, items from relevant scales (*Bartholomew et al., 2011*; *Liu & Chung, 2015*; *Longo et al., 2016*; *Rocchi et al., 2017*) and qualitative interviews. The combination of the two inputs is to maximize the content coverage of frustration of different basic psychological needs. Face validity of the items was examined by participants from the targeted population. Additionally, clarity, readability and comprehensibility of the items were checked. Content validity of the PNFS-PA was subsequently reviewed by experts in the SDT research filed. In the present research, the content validity index (CVI), the widely used index for scale evaluation as a quantitative approach (*Polit & Beck, 2006*), was computed for item retention or deletion decision, resulting in an initial scale of the PNFS-PA.

The initial scale was administered to participants drawn from the targeted population for further analysis. As suggested in previous research, the ESEM approach was introduced to examine the factorial validity of the scale (*Asparouhov & Muthén, 2009*; *Marsh et al., 2009*). Traditionally confirmatory factor analysis (CFA) was used in examining the factorial validity of a scale (*Mueller, 1996*). However, it has been criticized as relying on a highly restrictive independent cluster model. The model forces cross-loadings of items on unintended

**Table 4  Goodness-of-fit Indices of Models for Invariance Examination in Study 3 ($N = 652$).**

| | Model | $\chi^2$ | $df$ | CFI | TLI | RMSEA | 90% CI | SRMR | AIC | BIC | ABIC |
|---|---|---|---|---|---|---|---|---|---|---|---|
| Gender | Male ($N = 299$) | 106.081 | 52 | .976 | .959 | .059 | [.043, .075] | .021 | 12376.367 | 12624.297 | 12411.814 |
| | Female ($N = 353$) | 145.979 | 52 | .959 | .927 | .072 | [.058, .085] | .024 | 14771.497 | 15030.550 | 14817.999 |
| | M1 | 251.251 | 104 | .968 | .943 | .066 | [.056, .076] | .023 | 27147.864 | 27748.190 | 27322.741 |
| | M2 | 293.546 | 137 | .966 | .954 | .059 | [.050, .069] | .033 | 27134.123 | 27586.607 | 27265.933 |
| | M3 | 315.046 | 151 | .964 | .957 | .058 | [.049, .067] | .040 | 27124.480 | 27514.244 | 27238.020 |
| | M4 | 317.605 | 165 | .966 | .963 | .053 | [.044, .062] | .043 | 27119.838 | 27446.881 | 27215.107 |
| Age | G1 ($N = 326$) | 117.289 | 52 | .972 | .951 | .062 | [.047, .077] | .021 | 13664.622 | 13918.344 | 13705.824 |
| | G2 ($N = 326$) | 134.155 | 52 | .964 | .937 | .070 | [.055, .084] | .024 | 13468.446 | 13722.168 | 13509.649 |
| | M1 | 251.740 | 104 | .968 | .944 | .066 | [.056, .076] | .022 | 27133.068 | 27733.394 | 27307.945 |
| | M2 | 308.919 | 137 | .963 | .951 | .062 | [.053, .071] | .036 | 27136.215 | 27588.700 | 27268.025 |
| | M3 | 323.128 | 151 | .963 | .955 | .059 | [.050, .068] | .035 | 27117.022 | 27506.786 | 27230.562 |
| | M4 | 338.412 | 165 | .962 | .959 | .057 | [.048, .065] | .037 | 27128.451 | 27455.494 | 27223.720 |
| Independent samples | M1 | 236.559 | 104 | .971 | .949 | .063 | [.052, .073] | .022 | 27137.274 | 27737.600 | 27312.151 |
| | M2 | 290.444 | 137 | .967 | .956 | .059 | [.049, .068] | .037 | 27135.714 | 27588.198 | 27267.524 |
| | M3 | 306.226 | 151 | .966 | .959 | .056 | [.047, .065] | .040 | 27119.016 | 27508.780 | 27232.555 |
| | M4 | 313.107 | 165 | .968 | .964 | .052 | [.044, .061] | .037 | 27117.982 | 27445.026 | 27213.251 |

**Notes.**

Model fits to the two independent samples data are reported in the main text. Median age = 37.50 of the participants was used for group division (G1 and G2).

M1, configural invariance; M2, metric invariance (weak invariance); M3, scalar invariance (strong invariance); M4, item uniqueness invariance (strict invariance).
factors to zero, which overlooks the fact that the three factors in the scale are distinct yet related (*Deci & Ryan, 2014*). To overcome such limitation, ESEM is suggested to use to better represent the multi-dimensional structure of a scale, which allows items to freely cross-load on other factors (*Asparouhov & Muthén, 2009*). Therefore, we attempted to advance the literature by adopting a different statistical approach. The finalized 14-item scale demonstrated an adequate fit to the data via ESEM, supporting its factorial validity. Furthermore, the ESEM results from two independent samples (study 2 and 3) revealed that items loaded significantly higher in their intended factors rather than unintended factors, suggesting items in the newly developed scale explicitly measure a single need frustration, which is deemed an important item inclusion/exclusion criterion. In addition, the three factors' inter-correlations were better estimated in the ESEM, with a moderate correlation magnitude between three need frustration factors.

With regard to nomological validity of the scale, as expected, significant negative correlations between the three needs frustration and subjective vitality, and significantly positive correlations between the three needs frustration and negative affect were displayed. The findings substantiate SDT's argument and previous results (*Bartholomew et al., 2011*; *Vansteenkiste & Ryan, 2013*; *Liu & Chung, 2015*). The three needs frustration correlated more strongly with negative affect than subjective vitality, indicating it can be a better predictor for individuals' negative experiences under the PA context. Of note, in comparison with other contexts such as competitive sport (*Bartholomew et al., 2011*) and physical education (*Liu & Chung, 2015*), the correlation magnitude between needs frustration and well/ill-being measurements in the PA context was relatively lower. It may be ascribed to the specific nature of the PA context, which involves a higher degree of freedom and more self-directed choice compared with other contexts such as competitive sport and physical education which are more compulsory. As such, it is possible that people may opt-out or relapse from PA participation to prevent their basic needs from being frustrated. Another probable explanation is that, for the individuals targeted in the current research (general Chinese adults), a large proportion of them are physically inactive. Hence, PA engagement appears to be a less significant component in their daily lives, and the possible negative experiences from the PA context may generate a less critical impact on perception of their overall subjective well/ill-being. Therefore, the relatively low association intensity observed was considered to be reasonable. Nonetheless, given the health benefits of PA engagement along with the adverse influence of need frustration that can repel people from PA participation, need frustration should be of causation.

As to the reliability of the scale, multiple indicators were introduced in the present research. That is Cronbach's alpha coefficient, composite reliability and test-retest reliability. Overall, the result suggests that the newly developed scale holds sufficient internal consistency reliability and temporal stability. Moreover, measurement invariance (weak, strong, and strict) of the scale was appraised across gender, age and independent samples. Weak, strong and strict measurement invariance requires factor loadings, factor mean (reflected by invariant intercepts) and residual variance to be equivalent across groups, which constitutes progressively constrained models (*Meredith & Teresi, 2006*).

Findings on the measurement invariance of the PNFS-PA suggest that scores derived from different groups are comparable.

It is worthy to note that need frustration, which is the interest of the current research, differs from need thwarting. Need frustration can be described as a perception of basic psychological needs are thwarted, hence it is a subjective experience (*Bartholomew et al., 2011*). Conversely, need thwarting is the act of thwarting people's basic psychological needs (*Quested et al., 2018*). Past studies sometimes use the two terms interchangeably. That is, they measured need frustration while referring it as need thwarting (*Bartholomew et al., 2011*). To avoid confusion, the name of our scale explicitly indicates that need frustration is the target of measurement. Plus, items in the PNFS-PA explicitly refer to feelings related to need thwarting behaviors, rather than need thwarting behaviors themselves. By doing so, the scale avoids a possible risk of mixing two related but distinct concepts in SDT.

The current research has several implications. First, the newly developed scale makes a measure of need frustration available, which could assist researchers in further exploring need frustration experiences in the context of PA. Second, from a theoretical perspective, it provides empirical support to the basic psychological need theory, namely needs for autonomy, competence and relatedness in a specific context (i.e., PA context). Third, it offers further insight into the motivational roots of maladjustment from an SDT approach. As the findings highlight a negative association between need frustration and negative affect, the practical implication is that it is important to tackle the negative psychological experiences in the PA context.

Though promising evidence on the psychometric properties of the need frustration scale is obtained, limitations should be acknowledged. First, samples in the present study were drawn via a convenience sampling strategy which may not be representative of the target population (*Fogelman, 2002*). In order to strengthen the generalizability of the findings, future validation studies should attempt to recruit participants via random sampling that can better represent the population. Also, we adopted samples from study 2 and 3 for the multi-group ESEM analysis. While the analysis results corroborated the invariance of the instrument, it would be a better strategy to recruit two new samples in study 3 for such an analysis. Second, only three experts were involved in the content validity appraisal stage. Despite they had expert knowledge germane to the present research, it is still recommended that a larger number of experts in different areas be included (*Rubio et al., 2003*). Third, while the face validity and content validity of the PNFS-PA were checked before formal administration, a subsequent pilot study could have been desirable. Such a pilot study could be useful in troubleshooting any potential unforeseen issues prior to large-scale questionnaire administration (*Johanson & Brooks, 2010*). Fourth, regarding the nomological validity evaluation of the PNFS-PA, only subjective vitality and negative affect scales were employed. Despite the included constructs are suggested to be linked with need frustration and are important criteria for nomological validity assessment, other measures should also be taken into consideration. For instance, future research should consider entailing behavioral motivations of PA behavior for further validity assessment purpose. Within the framework of SDT, the frustration of the basic psychological needs is expected to exhibit a positive association with controlled behavioral motivation and a negative

association with autonomous behavioral motivation (*Ryan & Deci, 2017*). Moreover, some other alternative measures of maladaptation such as psychological distress should be included to further validate the newly developed scale in future studies.

## CONCLUSIONS

In summary, although scale validation is an on-going process, the current research developed and initially validated the PNFS-PA, exhibiting its psychometric solidness. The scale can be useful for addressing need frustration in the PA specific context. Investigation of need frustration in PA should be of importance, as need frustration is detrimental experience individuals may have when engaging in PA behavior. It can cause negative outcomes such as negative affect, mal-adaptation and subjective ill-being and PA relapse. Therefore, PA intervention practice is informed to redress it. Additionally, individuals' important others (e.g., friends, family members, PA instructors) should be better informed with the adverse consequences of their conscious/unconscious need thwarting behavior in the PA context through educational and interventional campaigns. Thus, need frustration experiences can be better curbed. In all, this research contributes to the literature by providing a psychometric sound need frustration instrument for use in the PA context, and it is hoped that the availability of such a valid and reliable scale can facilitate and promote pertinent research in the future.

### Funding

The research was funded by the Faculty Research Grant (FRG) of Hong Kong Baptist University (FRG1/16-17/039). The funders had no role in study design, data collection and analysis, decision to publish, or preparation of the manuscript.

### Grant Disclosures

The following grant information was disclosed by the authors:
Faculty Research Grant (FRG) of Hong Kong Baptist University: FRG1/16-17/039.

### Competing Interests

The authors declare there are no competing interests.

### Author Contributions

- Pak-Kwong Chung and Tao Zhong conceived and designed the experiments, performed the experiments, analyzed the data, prepared figures and/or tables, authored or reviewed drafts of the paper, and approved the final draft.
- Jing-Dong Liu conceived and designed the experiments, authored or reviewed drafts of the paper, and approved the final draft.
- Chun-Qing Zhang analyzed the data, authored or reviewed drafts of the paper, and approved the final draft.
- Ming Yu Claudia Wong analyzed the data, prepared figures and/or tables, and approved the final draft.

## PeerJ

## Human Ethics

The following information was supplied relating to ethical approvals (i.e., approving body and any reference numbers):

The current study involves human participants and the research protocol had been approved by the Committee of Research Ethics and Safety at the Hong Kong Baptist University (Project Code: HASC/16-17/0304).

## Data Availability

All raw data are available in the Supplemental Files.

## Supplemental Information

Supplemental information for this article can be found online at http://dx.doi.org/10.7717/peerj.9210#supplemental-information.

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
