# Peer review of "Development and initial validation of the Psychological Need Frustration Scale for Physical Activity"

_PeerJ, doi:10.7717/peerj.9210_

## Round 0.1 · original submission · Major Revisions

I now have received three reviewers' comments. Although all of them expressed their interest in your study, several aspects of this manuscript should be revised to improve its clarity. Their observations are presented with clarity so I'll not risk confusing matters by belaboring or reiterating their comments. While I might quibble with the occasional point, I note that I regard the reviewers' opinions as substantive and well-informed. I believe that all of the highlighted reservations require contemplation and appropriate attention in revising the document if it is to contribute appropriately to Peerj and the extant literature. Please revise or refute according to the three reviewers' comments and provide a point by point reply in addition to the revised manuscript.

Tsung-Min Hung, PhD., FNAK, FISSP
PeerJ editor
Research chair professor,
Department of Physical Education,
National Taiwan Normal University

Reviewer 1 ·

Basic reporting

The manuscript is written in a good level of English throughout.
Clear structure of the manuscript and the purpose of the study was clearly stated. The references cited were relevant and appropriate. Tables are easy to read though the decimals should be consisted throughout. Figure(s) could be used for illustrating the final outcome of the ESEM.

Experimental design

The rational was clearly stated for selecting the research method, although the process of forming the initial item pool could be further described. For example, the background of the three specialists in psychology and SDT, do they have the related knowledge of the competitive sport and physical education? More expert in different areas could be included for providing the evidence of validity.
Also, the sampling method should be further elaborated, where are the areas for recruiting the participants for forming the item pool? Was there a diversity geographically for sampling from the population? If these could be clearly stated one may be able to replicate the study in the future.

Validity of the findings

The results appear that the questionnaire has good validity and is reliable across the sample, though the necessity for developing the questionnaire was not clearly stated. What are the future implications of this questionnaire? Is this measurement beneficial for better understanding of the theory or can be used to increase our knowledge in the sport and physical education areas? All these points could be addressed further in the discussion section.

Additional comments

Line 266 "...were the same as study 2 and 3." should be "study 1 and 2" instead.
Line 173 “Results and Discussion” should be in bold.

Reviewer 2 ·

Basic reporting

The aim of the submitted paper entitled “Development and initial validation of the
Psychological Need Frustration Scale for Physical Activity” was to develop and validate a new scale based on self-determination theory (SDT). The background of the questions is clear and well structured. The literature is relevant and appropriate to address the questions and research gap.

Experimental design

1. Why the authors did not conduct a pilot study in study #2? This step is very important to confirm the process of the formal study will not go wrong? Please report it.
2. Table # 1 shows the basic information of study #2, why I did not see internal consistency coefficients of the subscales? Why study # 2 did not show internal consistency coefficients until study #3? I suppose this basic reliability index is very important. Also, I found the initial of the relatedness need frustration is not consistent, sometimes you use “RO” while sometimes you use “RE.”
3. In table 2, the item of COMPETENCE “ When engaging in physical activity, at times you feel…you feel inadequate in some situation.” Inadequate in what?
4. In study #3, you perform a multi-group ESEM, why you did not TWO new samples in study #3? Instead, you use one sample from study #2 and one sample from study #3, is there any rationale to support you to analyze by this approach?
5. Could you make a table for the criterion-related validity correlations among need frustration, subjective well-being, and negative affect? Also, in this table, you may show the internal consistency coefficients of all the scales that you used.
6. Line 404, you said your reliability including test-retest reliability? Really? In which study? How many participants for this study? And, what is the time period before the first test and second test?

Validity of the findings

The validity of the finding can be contributed to the researchers only when if this pape is well revised.

Additional comments

What is other potential construct validity index you may use but not available in this study? Could you suggest them for future research?

Reviewer 3 ·

Basic reporting

1)Gunnell et al(2013) have shown a clear definition of PA in that study (please see footnote 1), however in your paper PA only have been mentioned but no further information regarding its content. did all the participants familiar and understand? please explain what you have done regarding this matter.

2)the intro part of this manuscript is easy to read, but I would suggest that basic need thwarting content need to be added at line 52-70. especially what criteria should be included/excluded items.

3)line 61-64 need to re-exam the content of competence and relatedness need frustration.

Experimental design

1)study 1 need more detail for definition of PA, how and where PNFS-PA item pool was formed. I would suggest that Bartholomew et al(2011B) has done a nice work, it is a source to consult.
2)line 167-168 for the CVI part is not clear, suggest improve.
3)line 239, please double check two degree of freedom, are they all correct?

Validity of the findings

no comment

Additional comments

a psychometrically sound measurement is desirable in every aspect of the sport, exercise and pa research. however, a theoretical based and psychometric fit one is not an easy job. I commend the authors for their effort to make one of the scale for further research.

---

## Round 0.2 · accepted · Accept

I have now received all the reviewers’ comments indicating their satisfaction with your reply and revisions from previous comments. You and your co-authors have my congratulations. Thank you for choosing PeerJ as a venue for publishing your research work and I look forward to receiving more of your work in the future.

Tsung-Min Hung, PhD., FNAK, FISSP
PeerJ editor
Research chair professor,
Department of Physical Education,
National Taiwan Normal University

Reviewer 2 ·

Basic reporting

Yes, English writing is clear and unambiguous. The background of the questions is clear and well structured. The literature is relevant and appropriate to address the questions and research gap. The tables are appropriate and well-presented.

Experimental design

The authors have properly addressed my questions regarding experimental design, missing values in tables, inconsistent presentation of abbreviations of the variables, and inappropriate presentation of the texts. The revised manuscript looks better than the previous manuscript. Also, the authors addressed the limitation that they haven’t done a pilot study in the first beginning. By such advising, future studies might learn from their experiences. The authors also accepted my suggestions to redo tables for criterion validity and conduct a new study to test test-retest reliability.

Validity of the findings

after the revision of the contents, and recollecting the data and analyses, the validity of the findings are robust and meaningful.

Additional comments

After your efforts to revise, recollect data, and redo the tables, it is better than the previous manuscript. I accept this manuscript. Congratulations.

Reviewer 3 ·

Basic reporting

The manuscript is re-writing according to reviewers' suggestion in a good level of English throughout. overall structure is clear and sufficient in literature review. several suggestions have been taken into consideration and the quality of the manuscript has been imporved.

Experimental design

aims and scope of this manuscript now is consistent with the journal's

Validity of the findings

no more revision is needed.